# Liquid-Phase Exfoliated Silicon Nanosheets: Saturable Absorber for Solid-State Lasers

**DOI:** 10.3390/ma12020201

**Published:** 2019-01-09

**Authors:** Mengxia Wang, Fang Zhang, Zhengping Wang, Xinguang Xu

**Affiliations:** 1State Key Lab of Crystal Materials, Shandong University, Jinan 250100, China; mengxiawang1989@126.com (M.W.); xgxu@sdu.edu.cn (X.X.); 2School of Physics and Optoelectronic Engineering, Shandong University of Technology, Zibo 255000, China; yourszhangfang1988@126.com

**Keywords:** Si nanosheets, 2D material, passively Q-switched, solid-state laser

## Abstract

As a newly-developed two-dimensional (2D) material of group-IVA, few-layer silicon (Si) nanosheets were prepared by the liquid phase exfoliation (LPE) method. Its non-linear saturable adsorption property was investigated by 532 and 1064 nm nanosecond lasers. Using Si nanosheets as the saturable absorber (SA), passive Q-switched all-solid-state lasers were demonstrated for the first time. For different laser emissions of Nd^3+^ at 0.9, 1.06, and 1.34 µm, the narrowest Q-switched pulse widths were 200.2, 103.7, and 110.4 ns, corresponding to the highest peak powers of 2.76, 2.15, and 1.26 W. The results provide a promising SA for solid-state pulsed lasers and broaden the potential application range of Si nanosheets in ultrafast photonics and optoelectronics.

## 1. Introduction

The most famous 2D material, graphene, was exfoliated successfully from graphite for the first time in 2004 [1]. Since then, other 2D materials like graphene oxide [2], black phosphorus (BP) [3], topological insulators [4], and transition metal dichalcogenides (TMDs) [5] have been widely investigated and found to have potential applications in many fields such as biology, medical, communication and microelectronics. In the nonlinear optical (NLO) field, 2D materials have received increasing attention in recent years [6,7,8,9].

Passive Q-switching is the preferred technique to generate nanosecond and even picosecond laser pulses which possesses many advantages such as high performance, simplicity, and compactness. However, traditional saturable absorbers (SAs), like ion-doped crystals and semiconductor saturable absorber mirrors (SESAMs), are wavelength-sensitive and suffer from narrow modulation bandwidth. [10,11,12,13]. The emergence of graphene-like 2D materials accelerates the development of passively Q-switched lasers to an ultrabroad spectral region [5,14,15]. Compared with the traditional SAs, the 2D SAs have shown excellent saturable absorption properties including ultrafast recovery time, controllable modulation depth, easy fabrication, wide waveband absorption, as well as low cost. They can easily generate short laser pulses by compressing the laser sizes, which is superior to active Q-switching methods, e.g., acousto-optic [16] and electro-optic modulation [17].

There are still many problems in the practical applications of 2D SAs. For example, graphene possesses a low modulation depth and large non-saturable losses [14], BP is very unstable in ambient atmosphere and may degrade within a few hours, and topological insulators require complex preparation processes [18]. So the active search for more superior 2D SAs is still underway. The features of some typical SA materials are shown in Table 1 (CNT is carbon nanotubes). We have tried to find suitable 2D SA materials from the group IVA elements. Such as silicon (Si), germanium (Ge), and tin (Sn), which are isoelectronic to graphene. Among them, Si is one of the most important materials in modern electronic technology and its nanostructure has many applications due to size-confinement effect [19,20,21]. Si-based devices are easier to fabricate as Si is compatible with complementary metal-oxide-semiconductor (CMOS) technology. Extensive research has been done on the synthesis of silicon quantum dots (zero-dimensional), silicon nanowires (one-dimensional), and 2D allotropic forms [22,23,24]. At the same time, the nanostructure Si materials have been promising candidates as anodes of lithium-ion batteries, which can be used in highenergy density portable electronics, large-scale energy storage systems, and electrical vehicles [25,26].

Recently, there have been many reports on the synthesis of Si nanosheets. For example, high quality Si nanosheets were prepared from inexpensive natural clays via a one-step simultaneous molten salt-induced exfoliation and chemical reduction process [27]. Using graphene oxide sheets (GOs) as sacrificial templates, Lu et. al developed a facile method to synthesize free-standing ultrathin silicon nanosheets [28]. Amine-modified Si nanosheets were obtained by the exfoliation of layered polysilan. They are easily self-assembled into a regular stacking structure [29]. At present, research on the non-linear optical applications of Si nanosheets is still very scarce and its non-linear saturable properties were demonstrated in 2018 for the first time [30]. 

In this study, the saturable absorption response of Si nanosheets was investigated at 532 and 1064 nm by nanosecond lasers. The modulation depth and saturable intensity were measured to be 18.9% and 16.2 MW/cm^2^, 17% and 19.8 MW/cm^2^ for 532 and 1064 nm wavelength, respectively. Using Si nanosheet-coated quartz substrate as SA, the passively Q-switched operations for solid-state lasers were realized at 0.9, 1.06, and 1.34 µm with a compact concave-plane resonator.

## 2. Materials and Methods 

Liquid phase exfoliation (LPE) has been widely used to produce high-quality 2D nanosheets because of its effectiveness and straightforwardness. In this study, the few-layer Si nanosheets were prepared from bulk Si by the LPE method assisted with sonication. High purity ethanol was used as the organic solvent for the feature of hypotoxicity. In the process of exfoliation, no surfactant was required due to the on-surface isolation. Firstly, a commercial high-purity bulk Si crystal (purity 99.999%) was milled in an agate mortar for 30 min, and the obtained Si powder was dissolved in absolute ethanol. The mixed liquid was ultra-sonicated for 2 h. The dispersion was then settled for 3 days to deposit undispersed particles. The final supernatant liquor was the needed Si nanosheets dispersion (Figure 1a), which was directly used for the saturable absorption property measurement. For all other measurements in this paper, the Si nanosheet dispersion was dropped on the quartz substrate, and the dried component acted as the experimental sample (Figure 1a).

The morphologies of Si crystal powder and Si nanosheets were observed by a field-emission scanning electron microscope (FE-SEM, S-4800, Hitachi, Kyoto, Japan). As shown in Figure 1b, the Si powder displayed representative solid agglomerates with typical dimensions of several micrometers, which formed a stacked layered structure. The cleavage steps and cleavage planes manifested the weak binding force among layers. The SEM image of Si nanosheets were shown in Figure 1c. The nanosheets exhibited laminar morphology, which was in stark contrast to the appearance of bulk Si. Meanwhile, the Si nanosheets are almost transparent because of their ultrathin nature. The thickness of Si nanosheets was measured by atomic force microscopy (AFM, MultiMode Шa, Agilent, California, America). The Si nanosheets dispersion was dropped onto a sapphire substrate and settled in atmosphere for 24 h before measurement. Figure 1d,e show a typical AFM image and the corresponding height profile of Si nanosheets. The thickness was in the range of 4.67–5.58 nm, and the average thickness was 5.28 nm. Considering the single layer thickness of Si nanosheets is 0.31 nm [13], our sample is 17 layers.

To further confirm that the prepared sample was few-layer Si nanosheets, the Raman spectra (excitation wavelength: 633 nm, LabRAM HR800, HORIBA Ltd., Kyoto, Japan) of bulk Si powder and Si nanosheets were measured. The detailed results are shown in Figure 1f. For the bulk Si powder obtained from bulk material, there was one sharp Raman peak at 522.6 cm^−1^ originating from the microcrystalline [31]. For the Si nanosheets, the Raman peak appeared at 511.8 cm^−1^, which presented an obvious red shift of 10.8 cm^−1^ with a broadened FWHM (full width at half maximum), compared to the Si powder. Such properties of Si nanosheets have been predicted in theory and proved by different experiments [27,28,32,33,34], and our measurement results are consistent with those obtained by J. Ryu et. al [27], where a 11 cm^−1^ red shift to 508 cm^−1^ observed from the 5 nm thickness Si nanosheets. Meanwhile, the commonly observed peaks corresponding to amorphous surface oxides between 300 and 450 cm^−1^ as well as the peak at 480 cm^−1^ appearing in amorphous Si were not found in our measurement.

## 3. Results and Discussion

### 3.1. Saturable Absorption Performance

A 10 Hz, nanoseconds electro-optic Q-switched Nd:YAG laser (self-built) was used as the light source to characterize the saturable absorption property of Si nanosheets by measuring its transmittance variation with the incident power density. The pulse durations of fundamental wave (1064 nm) and its frequency doubling wave (532 nm) were 19, 16 ns, respectively. The as-prepared Si nanosheets dispersion was held in a quartz cuvette with 1 mm thickness. As a control test, a quartz cuvette with solvent alcohol was measured at different energy levels and no NLO effect was observed. So the affection of cuvette and alcohol on the NLO response signal can be neglected. The experimental results are shown in Figure 2.

The power-dependent transmittance can be fitted by the following equation [35]:(1)$$T=A\exp\left(\frac{-\delta T}{1+\frac{I}{I_{S}}}\right)$$
where *A* is a normalization constant, *δT* is the absolute modulation depth, *I* is the incident intensity, and *I*_S_ is the saturation intensity. The fitted absolute modulation depths are 18.9% at 532 nm and 17.0% at 1064 nm. Considering the initial transmittance of Si nanosheets dispersion were 77% (532 nm) and 80% (1064 nm), the ultimate transmittance can reach about 96% in theory. The saturation intensity *I*_S_ were fitted to be 16.2 MW/cm^2^ for 532 nm and 19.8 MW/cm^2^ for 1064 nm, respectively.

### 3.2. Passively Q-Switched Operation

Using Si nanosheet-coated quartz substrate as SA, passively Q-switched lasers were operated with Nd^3+^ doped crystals as gain materials at 0.9, 1.06, and 1.34 µm wavelengths. Similar to our previous report [36], a plano-concave laser resonator was used in this work as shown in Figure 3. An 808 nm fiber-coupled diode laser with the core diameter of 100 µm was used as pump power. The pump light was delivered into the Nd^3+^ laser crystal by a 1:1 imaging module. A Nd:YAG crystal was used as gain material with the dimensions of 4 × 4 × 7 mm^3^ for 0.9 and 1.06 μm laser operations. Both surfaces of this crystal were anti-reflection (AR) coated at 0.9–1.1 μm waveband. For 1.34 μm emission, the gain medium was a Nd:YVO_4_ crystal with the dimensions of 3 × 3 × 5 mm^3^. To avoid the prior oscillation of 1.06 μm laser, its transmission surfaces were both AR coated at 1.06 and 1.34 μm. M1 was a flat mirror which AR coated at 808 nm, and high-reflection (HR) coated at working wavelengths. M2 was a concave mirror and partially transmitted at working wavelengths. Under our experimental abilities, the M2 parameters were respectively optimized at each wavelength: for the 0.9 µm laser operation, its transmittance T = 5%, the curvature radius R = 50 mm. For 1.06 μm, T = 10%, R = 100 mm. For 1.34 μm, T = 5%, R = 100 mm. The cavity length between M1 and M2 was 27 mm for the 0.9 µm laser, 24 mm for 1.06 μm laser and 27 mm for 1.34 μm laser, respectively. The temporal behaviors of the passively Q-switched laser were recorded with a digital oscilloscope (DPO7104, Tektronix Inc., Beaverton, OR, America). 

The continuous-wave (CW) laser output was obtained when the Si nanosheet SA was removed from the resonator. The optical conversion efficiencies were 34.1%, 36.1% and 42.8% for 0.9, 1.06 and 1.34 μm, respectively. When the Si nanosheet SA was inserted into the laser cavity and the pump power increased, the stable passively Q-switched operation could be achieved. The detailed results are summarized in Figure 4 and Figure 5.

The threshold pump power for the passively Q-switched operations were 2.41, 1.10, and 0.27 W for 0.94, 1.06, and 1.34 μm operations, respectively. As shown in Figure 4a–c, the Q-switched average output power and peak power increased linearly with the absorbed pump power for three wavelengths. To prevent the Si nanosheets SA from being damaged and lowering the thermally induced loss in the laser crystal and SA, the pump power was limited in a certain range. The stable passively Q-switched operations could be maintained until the absorbed pump power increased to 4.52, 3.35, and 1.47 W for 0.94, 1.06, and 1.34 μm operations, respectively. The achieved maximum Q-switched average output powers of 0.94, 1.06, and 1.34 μm were 164, 131, and 79 mW, corresponding to the absorbed pump powers of 4.5, 3.35, and 1.47 W, respectively. The highest peak powers were 2.76, 2.15, and 1.26 W for 0.94, 1.06, and 1.34 μm, respectively. 

With the absorbed pump power elevated, the time interval between the adjacent pulses and the pulse durations decreased for three wavelengths. During the 0.9 μm laser operation, in the absorbed pump power region of 2.41–4.52 W the pulse width decreased from 403 to 200.2 ns, and the corresponding repetition frequency increased from 166.6 to 294.5 kHz. For 1.06 μm operation, the pulse width decreased from 248.6 to 103.7 ns, and the repetition frequency increased from 319.8 to 587 kHz in the absorbed pump power range of 1.10–3.35 W. For the 1.34 μm operation, the pulse width varied from 296 to 110.4 ns with the repetition frequency monotonously increased from 238.5 to 570 kHz in the absorbed pump power region of 0.27-1.47 W. The largest pulse energies calculated were 0.55, 0.22, and 0.14 μJ for 0.9, 1.06, and 1.34 μm, respectively.

The typical pulse behaviors at three wavelengths are demonstrated in Figure 4d–f. The tidy and clean pulse sequences indicated the good results of Si-based Q-switched lasers. When the pump power was increased further, the pulse performance became degraded and the satellite pulses were found in the neat pulse sequence. If the pump power continued to be increased, the Q-switched pulse trains would quickly turned into CW operation. The stable Q-switched pulse trains would reappear if the pump power were adjusted back to the previous range. This indicated the Si SA was not damaged in the Q-switching process.

Figure 5a–c show the emission spectrum of passively Q-switched lasers; the center wavelengths were located at 946.4, 1064.3, and 1341.8 nm for 0.9, 1.06, and 1.34 μm operations, respectively.

The Q-switching conversion efficiencies with respect to the CW operiation were 10.6%, 10.8%, and 12.5% for 0.9, 1.06, and 1.34 μm lasers, respectively. The results revealed the notable non-saturable loss of the present Si-based SA component. On account of the SA substrate being an uncoated quartz glass, which has large absolutely non-saturable linear loss, the insertion loss of the Q-switch mode has been appreciably increased. Therefore, the low Q-switching conversion efficiency with respect to the CW mode were mainly attributed to the uncoated quartz substrate, not the Si nanosheet itself.

The detailed output characteristics of Si-based Q-switched lasers at three wavelegths are summarized in Table 2 with some recent experimental results about common 2D materials (graphene, MoS_2_, antimonene (Sb), Bi_2_Se_3_, ReS_2_) based Q-switched lasers as comparisons (*λ* is the wavelength, LC is the laser crystal, *τ* is the pulse width, *f* is the repetition rate, SP is the single pulse energy, PP is the peak power). When the laser crystals are of similar Nd^3+^ ions doped crystals, among the listed 2D materials in Table 1, the Si-based laser gives a narrower pulse width and higher peak power than MoS_2_ and Sb, but is inferior to graphene for the 0.9 μm laser operation. For 1.06 μm laser operation, the pulse width of Si nanosheets is narrower than graphene, MoS_2_, and Bi_2_Se_3_. In addition, the peak power of Si is higher than MoS_2_ and Bi_2_Se_3_, but lower than graphene. For the 1.34 μm laser operation, Si nanosheets give the narrowest pulse widths among four kinds of materials. But the pulse peak power was lower than graphene and Bi_2_Se_3_. In summary, compared with other typical 2D materials, Si nanosheets have exhibited comparable passively Q-switching properties under similar conditions like the gain medium and output wavelength.

## 4. Conclusions

The saturable adsorption property of Si nanosheets was demonstrated by nanosecond lasers. With Si nanosheet-coated quartz substrate as SA, its passively Q-switched operations for Nd^3+^ solid-state lasers were successfully realized for the first time. AFM measurement revealed that the thickness of our Si nanosheet sample, which was prepared by the LEP method, was 5.28 nm, i.e., 17 layers. This attribute was further verified by a Raman spectrum. At three different Nd^3+^ laser wavelengths of 0.9, 1.06, and 1.34 µm, the shortest pulse widths were 200.2, 103.7, 110.4 ns, and the largest single pulse energies were 0.55, 0.22, 0.14 μJ, respectively. This work shows that Si nanosheets can be used as effective optical modulating devices for a solid-state laser. As a low-cost, high-yield 2D material, the stable, broadband NLO response will help Si nanosheets to find new applications in optics and optoelectronics.

## Figures and Tables

**Figure 1 materials-12-00201-f001:**
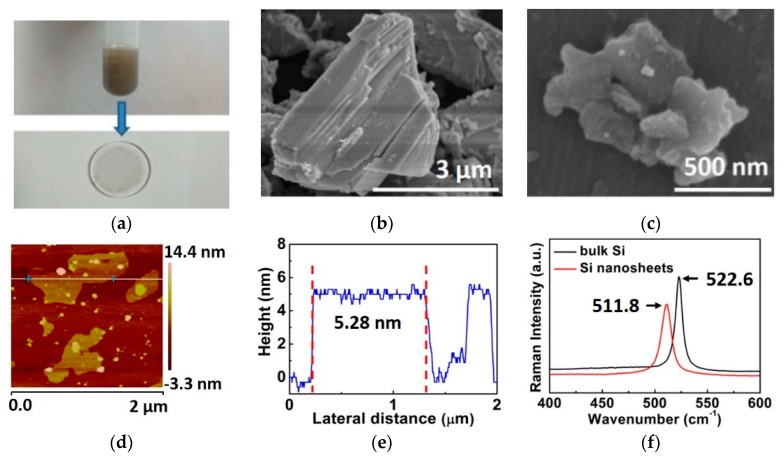
(**a**) Photographs of the Si nanosheet dispersion and the Si nanosheets coated on quartz substrate. (**b**) Scanning electron microscope (SEM) image of Si powder and (**c**) SEM image of Si nanosheets. (**d**,**e**) Atomic force micrsocope (AFM) image and corresponding height profile of Si nanosheets. (**f**) Raman spectra of bulk Si and Si nanosheets.

**Figure 2 materials-12-00201-f002:**
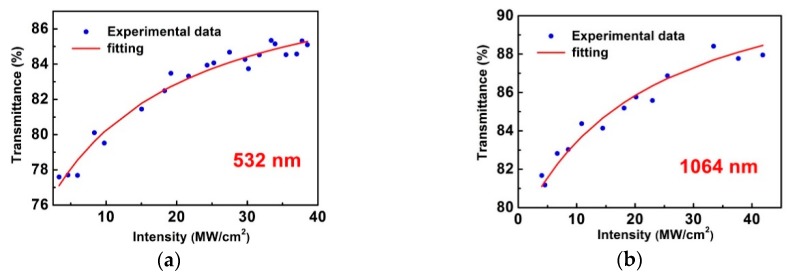
Transmittance versus incident power intensity for Si nanosheets dispersion at (**a**) 532 and (**b**) 1064 nm.

**Figure 3 materials-12-00201-f003:**
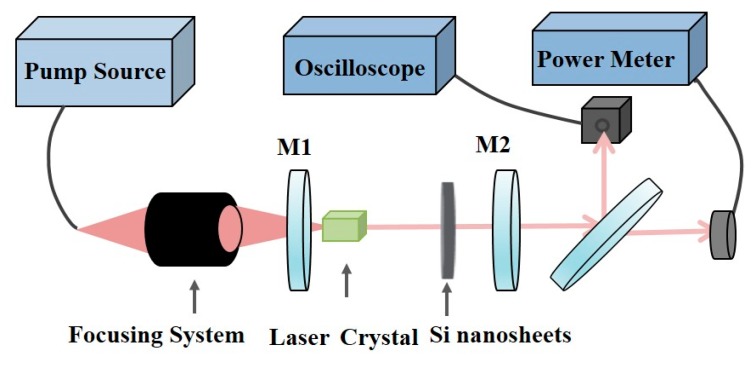
Experimental set-up of the Si nanosheets Q-switched solid-state laser.

**Figure 4 materials-12-00201-f004:**
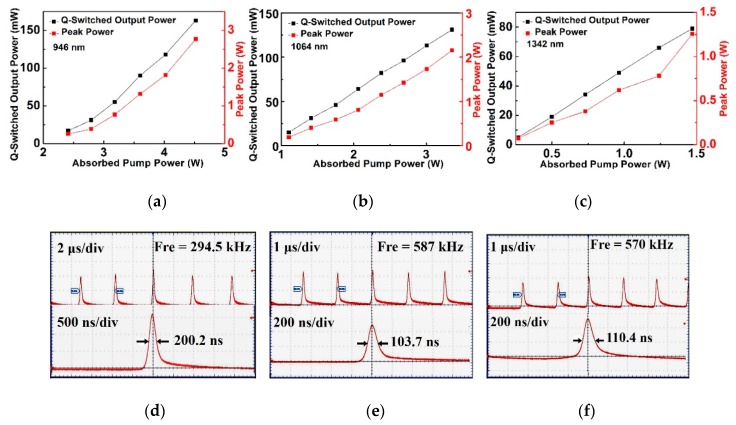
The detailed output characteristics of Si nanosheets based Q-switched lasers. (**a**–**c**) The maximum Q-switched average output power and highest peak power at 0.9, 1.06, and 1.3 μm, respectively. (**d**–**f**) The typical pulse trains at 0.9, 1.06, and 1.3 μm, respectively.

**Figure 5 materials-12-00201-f005:**
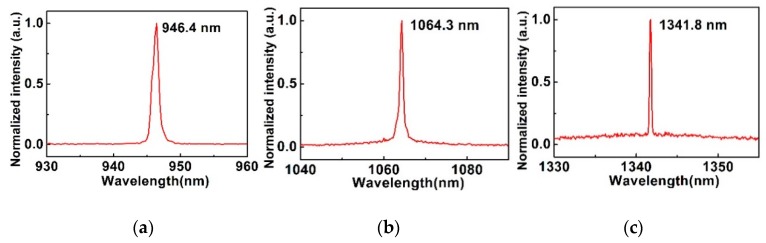
The output spectra of Si nanosheets based Q-switched lasers. at (**a**) 0.9, (**b**) 1.06, and (**c**) 1.3 μm.

**Table 1 materials-12-00201-t001:** The pros and cons of some typical saturable absorber (SA) materials.

Materials	Advantages	Disadvantages
Semiconductor saturable absorber mirror (SESAM)	Mature technology; Wide applications	Complex fabrication; Narrow wavelength range; Recovery time and modulation depth are difficult to adjust; High cost
Graphene	Zero band gap and no selectivity for wavelength range; Simple fabrication; Low cost	Low modulation depth; Large nonsaturable losses
Carbon nanotube (CNT)	Wide waveband absorption; Low cost	The absorption wavelength range is determined by the diameter of the tube; Difficult to disperse
Black phosphorus (BP)	Direct and layer-sensitive bandgap	Unstable in ambient atmosphere
Tungsten disulfide(WS_2_)	Layer-sensitive bandgap	Complex fabrication

**Table 2 materials-12-00201-t002:** Passively Q-switching performance for solid-state laser of Si nanosheets and some typical 2D materials.

*λ* (μm)	Material	LC	Laser Properties				Ref
			*τ* (ns)	*f* (kHz)	SP (μJ)	PP (W)	
0.9	Si	Nd:YAG	200.2	294.5	0.55	2.76	this work
	graphene	Nd:YAG	162	225	4	24.7	[37]
	MoS_2_	Nd:YAG	280	609	0.35	1.23	[38]
	Sb	Nd:YAG	208.8	268.3	0.31	1.48	[36]
1.06	Si	Nd:YAG	103.7	587	0.22	2.15	this work
	graphene	Nd:YAG	260	167	8.32	32	[39]
	MoS_2_	Nd:GdVO_4_	970	732	0.31	0.32	[40]
	Bi_2_Se_3_	Nd:GdVO_4_	666	547	0.0585	0.09	[41]
1.3	Si	Nd:YVO_4_	110.4	570	0.14	1.26	this work
	graphene	Nd:GdVO_4_	450	43	2.5	5.56	[42]
	ReS_2_	Nd:YAG	403	214	0.42	0.9	[43]
	Bi_2_Se_3_	Nd:YLF	433	161.3	1.23	2.84	[44]

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
