# Peer review of "Liquid-Phase Exfoliated Silicon Nanosheets: Saturable Absorber for Solid-State Lasers"

_materials, 2019, doi:10.3390/ma12020201_

Round 1

Reviewer 1 Report

The introduction section consists of some vague terms, long sentences. To assist readers, the long sentences can be cut short and written more clearly. Below I provide the example of some of the vague sentences as:

For the practical applications of 2D SAs, there are still some obstacles that need to be overcome

Recently, many studies have been focused…..

we turned our attention to …..

All materials are important in their respective field. Just because the author would like to use Si for their study, does not mean it is “the most important” material out of 114 metals in the periodic table, Right?  “Among them, Si is the most important material in modern electronic technology, and its nanostructure has many applications due to size-confinement effect [19].” 

Rewrite it as Si is one of the most important …….. Si-based devices are easier to fabricate as Si is compatible to the CMOS technology.

“the researches on the optical applications of Si nanosheets are still very scarce.” This sentence is untrue. Si has a long trend of being used for optical applications. Si has already been used as optical switches, thermal sensors, and biosensors. See the ref below and the reference therein:

"Compact Si-based asymmetric MZI waveguide on SOI as a thermo-optical switch." Optics Communications 410 (2018): 947-955. https://doi.org/10.1016/j.optcom.2017.10.007

To avoid the confusion to readers, authors may wish to combine the following sentence into one:

By time now the researches on the optical applications of Si nanosheets are still very scarce. As we have known, its nonlinear optical properties were demonstrated in 2018 for the first time.

Page 2 Line 58: The paper consists of many unnecessary terminologies and words that make the paper weak.  Break the sentence below into two or revise it:

 “In this work, the saturable absorption response of few-layer Si nanosheets was investigated by nanosecond lasers, the modulation depths and saturable intensities were determined to be 18.9% and 17.0%, 16.2 and 19.8 MW/cm2 60 for 532 and 1064 nm, respectively.”

Page 2 Lin 64: Define LPE when first used in the paragraph.

For figures, increase the font size of the x and y-axis titles.

Page 4 Line 142 Fig. 3: Does this circuit not come with a detector? Where is the light collected and by what means? Please enlarge the figure.

Fig 4 & 5: Visibility of the figure is poor. Label both x-and y-axes and increase the font sizes

Conclusion: 

Calculate the exact thickness of the fabricated Si here and provide that number instead of providing an approximate number.  Revise the sentence” The AFM measurement revealed that the thickness of our Si nanosheets sample, which was prepared by the LEP method, was about 5 nm, i.e. 17 layers or so”. Why about 5 nm? Provide the exact number here, e.g., 5.1 or 4.9, following the calculation.

Author Response

Thank you very much for your clear, detailed, and helpful comments, as well as so many specific suggestions. The related revisions are marked with red fonts in the revised manuscript.

Point 1: The introduction section consists of some vague terms, long sentences. To assist readers, the long sentences can be cut short and written more clearly. Below I provide the example of some of the vague sentences as:

a. For the practical applications of 2D SAs, there are still some obstacles that need to be overcome.

b. Recently, many studies have been focused…..

c. we turned our attention to …..

Response 1: We are sorry about the unclear expression in the introduction section, and we have improved it in the revised manuscript.

a. Page 1 Line 38: There are still many problems in the practical applications of 2D SAs.

b. Page 2 Line 52: Recently, there have been many reports on the synthesis of Si nanosheets. For example, high quality Si nanosheets were prepared from inexpensive natural clays via a one-step simultaneous molten salt-induced exfoliation and chemical reduction process [25].

c. Page 1-2 Lines 42-44: We have tried to find suitable 2D SA material from the group-IVA elements. Such as silicon (Si), germanium (Ge), and tin (Sn), which are iso­electronic to graphene. (in red)

Point 2: All materials are important in their respective field. Just because the author would like to use Si for their study, does not mean it is “the most important” material out of 114 metals in the periodic table, Right?  “Among them, Si is the most important material in modern electronic technology, and its nanostructure has many applications due to size-confinement effect [19].” 

a. Rewrite it as Si is one of the most important …….. Si-based devices are easier to fabricate as Si is compatible to the CMOS technology.

b. “the researches on the optical applications of Si nanosheets are still very scarce.” This sentence is untrue. Si has a long trend of being used for optical applications. Si has already been used as optical switches, thermal sensors, and biosensors. See the ref below and the reference therein:

"Compact Si-based asymmetric MZI waveguide on SOI as a thermo-optical switch." Optics Communications 410 (2018): 947-955. https://doi.org/10.1016/j.optcom.2017.10.007

To avoid the confusion to readers, authors may wish to combine the following sentence into one:

By time now the researches on the optical applications of Si nanosheets are still very scarce. As we have known, its nonlinear optical properties were demonstrated in 2018 for the first time.

Response 2: Thanks for your advice, we have rewrite the relevant sentences in the revised manuscript.

a. Page 2, Lines 44-47: Si is one of the most important materials in modern electronic technology and its nanostructure has many applications due to size-confinement effect [19]. Si-based devices are easier to fabricate as Si is compatible to the complementary metal-oxide-semiconductor (CMOS) technology.

b. Page 2, Lines 57-59: But the researches on the nonlinear optical applications of Si nanosheets are still very scarce and its nonlinear saturable properties were demonstrated in 2018 for the first time. (in red)

Point 3: Page 2 Line 58: The paper consists of many unnecessary terminologies and words that make the paper weak. Break the sentence below into two or revise it:

 “In this work, the saturable absorption response of few-layer Si nanosheets was investigated by nanosecond lasers, the modulation depths and saturable intensities were determined to be 18.9% and 17.0%, 16.2 and 19.8 MW/cm2 60 for 532 and 1064 nm, respectively.”

Response 3: OK. Page 2 Lines 60-62: In this work, the saturable absorption response of Si nanosheets were investigated at 532 and 1064 nm by nanosecond lasers. The values of modulation depth and saturable intensity were determined to be 18.9% and 16.2 MW/cm2, 17% and 19.8 MW/cm2 for 532 and 1064 nm, respectively. (in red)

Point 4: Page 2 Lin 64a. Define LPE when first used in the paragraph.

b. For figures, increase the font size of the x and y-axis titles.

Response 4: a. We defined “liquid phase exfoliation” as “LPE” in the abstract.

b. OK. (in red)

Point 5: Page 4 Line 142 Fig. 3: Does this circuit not come with a detector? Where is the light collected and by what means? Please enlarge the figure.

Response 5: The average output power was measured by a power meter ((Powermax 500D, Molectron Inc.), and the temporal behaviors of laser pulses were recorded by a digital oscilloscope (DPO7104, Tektronix Inc.) with a photodiode detector.

We redrew Figure 4 to make it clearer in the Page 5 Line 145. (in red)

Point 6: Fig 4 & 5: Visibility of the figure is poor. Label both x-and y-axes and increase the font sizes.

Response 6: OK.  (in red)

Point 7: Conclusion: 

Calculate the exact thickness of the fabricated Si here and provide that number instead of providing an approximate number.  Revise the sentence” The AFM measurement revealed that the thickness of our Si nanosheets sample, which was prepared by the LEP method, was about 5 nm, i.e. 17 layers or so”. Why about 5 nm? Provide the exact number here, e.g., 5.1 or 4.9, following the calculation.

Response 7: OK. Form the AFM image and the corresponding height profile of Si nanosheets (Figures 1d-e). We can get the thickness is within the range of 4.67 - 5.58 nm, and the average thickness is 5.28 nm. Considering the single layer thickness of Si nanosheets is 0.31 nm, our sample is 17 layers. (in red)

Reviewer 2 Report

The manuscript “Liquid-phase Exfoliated Silicon Nanosheets: Saturable Absorber for Solid-state Lasers” by Mengxia Wang et al. deals with the demonstration of silicon nanosheets as saturable absorber for different Nd:YAG laser wavelengths. The content of the manuscript is clearly presented and the topic is interesting for publication in Materials.

Several minor issues can be improved in the manuscript.

1. Lines 28-37 introduce different types of saturable absorbers. Maybe the section can be emphasized by an illustration or table showing different materials and types of SA with their pros and cons (wavelength range, bandwidth, suitable for mode-locking or q-switching, fabrication, applications etc). Since SAs are a wide and rapid-developing field there is a high risk on missing important references. An illustration or a table might give a better overview of the field of research and can be used to guide to the SA presented in the paper.

2. Line 53: GO is not explained.

3. Figure 5: Why is the bandwidth of the presented spectra so different? There should be a y-axis e.g. intensity or normalized intensity.

4. Table 1: 0.9 µm is missing. The different wavelengths should be divided by lines.

5. The achieved results should be discussed in more detail. When is Si better compared to graphene and why? What can be improved in the production of the SA to get better results?

6. Reference 43: 2015 should be bold.

Author Response

Thank you very much for your helpful comments, as well as detailed suggestions. The related revisions are marked with blue fonts in the revised manuscript.

Point 1: Lines 28-37 introduce different types of saturable absorbers. Maybe the section can be emphasized by an illustration or table showing different materials and types of SA with their pros and cons (wavelength range, bandwidth, suitable for mode-locking or q-switching, fabrication, applications etc). Since SAs are a wide and rapid-developing field there is a high risk on missing important references. An illustration or a table might give a better overview of the field of research and can be used to guide to the SA presented in the paper.

Response 1:  At present, a variety of materials have been found suitable for use as SAs. We have summarized the passively Q-switching performance for solid-state laser of Si nanosheets and some typical SA materials in Table 2. In order to avoid repetition, according to the suggestion of reviewer, we qualitatively compare the advantages and disadvantages of some typical SA materials in Table 1 (Line 65). (in blue)

Point 2: Line 53: GO is not explained.

Response 2: OK. We define “graphene oxide sheets” as “GOs” in line 55. (in blue)

Point 3: Figure 5: Why is the bandwidth of the presented spectra so different? There should be a y-axis e.g. intensity or normalized intensity.

Response 3: OK. We modified Figure 5 to make it easier to understand. (in blue)

Point 4: Table 1: 0.9 µm is missing. The different wavelengths should be divided by lines.

Response 4: OK. (in blue)

Point 5: a. The achieved results should be discussed in more detail.

b. When is Si better compared to graphene and why?

c. What can be improved in the production of the SA to get better results?

Response 5:  a. Page 5, Lines 152-177: We are sorry about the not detailed analysis in the results and discussion section, and we have improved it in the revised manuscript.

b. Page 6, Lines 202-209: When the laser crystal are similar Nd3+ ions doped crystals, the Si based laser gives narrower pulse widths and lower pulse peak power than graphene for 1.06 and 1.34 μm laser operations. But the passively Q-switching properties of Si based laser was better than Sb and MoS2 in Table 2. What we wanted to emphasize is that Si has exhibited comparable passively Q-switching properties with other typical 2D nanomaterials under the similar conditions like the gain medium and output wavelength. We are sorry for the misunderstanding caused by improper expression. We have corrected it in the revised manuscript. (in blue)

c. In this work, we choose LPE method to produce 2D Si nanosheets for its advantages of fast and straightforward. At the same time, the absolute ethanol was used as solvent for the effects of hypotoxicity. In order to obtain higher quality Si nanosheets dispertion, the surfactant such as polyvinyl alcohol solution can be added into solvent and prevent the reaggregation of the exfoliated Si nanosheets. In addition, the micromechanical exfoliation method and chemical vapor deposition method can also be considered for the preparation of Si nanosheets. Further, the Si nanosheets dispersion was dropped on quartz substrate in this report, the spin coating method can be used to overcome the shortcoming of an uneven coating

Point 6: Reference 43: 2015 should be bold.

Response 6:  OK. (in blue)

Round 2

Reviewer 1 Report

Authors have tried to address most of the questions asked in the first stage of review, except a few of them. They should carefully address all the comments provided to them to improve the quality of their manuscript and I list them below.

Page 2 Line 44 -47: “Si is one of the most important materials in modern electronic technology and its nanostructure has many applications due to size-confinement effect [19]. Si-based devices are easier to fabricate as Si is compatible with the complementary metal-oxide-semiconductor (CMOS) technology."

In addition to Ref [19], authors should consider citing relevant citations as:

1) Compact Si-based asymmetric MZI waveguide on SOI as a thermo-optical switch." Optics Communications 410 (2018): 947-955” https://doi.org/10.1016/j.optcom.2017.10.007

2) Chrostowski, Luka and Michael Hochberg. Silicon photonics design: from devices to systems. Cambridge University Press, 2015. http://cds.cern.ch/record/2017723

The manuscript contains too many grammatical errors in it. Careful proofreading is a must. 

Author Response

Thank you very much for your kind helps in the review process. We have carefully revised the manuscript for the second time. We hope the present version can satisfy the requirements of Mterials. The related revisions are marked with red fonts in the revised manuscript.

Point 1: Page 2 Line 44 -47: “Si is one of the most important materials in modern electronic technology and its nanostructure has many applications due to size-confinement effect [19]. Si-based devices are easier to fabricate as Si is compatible with the complementary metal-oxide-semiconductor (CMOS) technology."

In addition to Ref [19], authors should consider citing relevant citations as:

1) Compact Si-based asymmetric MZI waveguide on SOI as a thermo-optical switch." Optics Communications 410 (2018): 947-955” https://doi.org/10.1016/j.optcom.2017.10.007

2) Chrostowski, Luka and Michael Hochberg. Silicon photonics design: from devices to systems. Cambridge University Press, 2015. http://cds.cern.ch/record/2017723

Response 1: Thanks for the suggestion. We are delighted to cite these articles, and they are added in the Introduction part.

Point 2: The manuscript contains too many grammatical errors in it. Careful proofreading is a must. 

Response 2: We are very for the poor English writing, and we have improved it in the revised manuscript.
